# PLGA Nanoencapsulation Enhances Immunogenicity of Heat-Shocked Melanoma Tumor Cell Lysates

**DOI:** 10.3390/cells14241939

**Published:** 2025-12-06

**Authors:** Kevin Calderón Matheu, Benjamín Cáceres Araya, Fiorella Tarkowski Diaz, Natalia Hassan, Flavio Salazar-Onfray, Andrés Tittarelli

**Affiliations:** 1Instituto Universitario de Investigación y Desarrollo Tecnológico (IDT), Universidad Tecnológica Metropolitana, Santiago 8940577, Chile; kcalderon@utem.cl (K.C.M.); bcaceres@utem.cl (B.C.A.); ftarkowski@utem.cl (F.T.D.); nhassan@utem.cl (N.H.); 2Institute of Biomedical Sciences, Faculty of Medicine, Universidad de Chile, Santiago 8380453, Chile; fsalazar@uchile.cl; 3Millennium Institute for Immunology and Immunotherapy, Faculty of Medicine, Universidad de Chile, Santiago 8380453, Chile; 4Department for Basic and Clinical Oncology, Faculty of Medicine, Universidad de Chile, Santiago 8380453, Chile; 5Science for Life Laboratory, Department of Medicine, Karolinska Institutet, 17176 Stockholm, Sweden

**Keywords:** melanoma, tumor lysate, nanoparticles, PLGA, immunogenicity

## Abstract

**Highlights:**

**What are the main findings?**
PLGA nanoencapsulation preserves and significantly enhances the immunogenic activity of TRIMEL, a clinically validated heat-shock melanoma cell lysate used in TAPCells and TRIMELVax.NP-TRIMEL maintains functional stability for at least 24 weeks at 4 °C, enabling efficient TAPCells differentiation and generation of cytotoxic lymphocytes with >100-fold dose-normalized potency compared with soluble TRIMEL.

**What are the implications of the main findings?**
Stabilizing TRIMEL at 4 °C overcomes an important logistical barrier limiting large-scale deployment of TAPCells/TRIMELVax-like vaccines.Nanoencapsulation represents a feasible translational path to next-generation whole-tumor-cell vaccines, enabling improved accessibility, dose-sparing, and broader clinical applicability.

**Abstract:**

Therapeutic cancer vaccines have emerged as promising immunotherapy approaches. TRIMEL, a heat-shocked lysate derived from three melanoma cell lines, constitutes the basis of TAPCells and TRIMELVax cancer vaccines, both of which have shown clinical efficacy but face major limitations in stability and logistics due to the requirement of ultra-low temperature storage. In this study, we evaluated the encapsulation of TRIMEL into poly(lactic-co-glycolic acid) (PLGA) nanoparticles (NP-TRIMEL) as a strategy to enhance stability and preserve immunogenic function under more feasible storage conditions. NP-TRIMEL was synthesized using a double-emulsion method and characterized by hydrodynamic size, zeta potential, morphology, and TRIMEL loading. Functional assays using melanoma patient-derived monocytes and peripheral blood lymphocytes suggested that NP-TRIMEL promoted the generation of TAPCells capable of inducing cytotoxic lymphocytes against allogeneic melanoma cells, even after 24 weeks of storage at 4 °C. Remarkably, NP-TRIMEL showed a two-order-of-magnitude increase in efficiency compared to the original TRIMEL in promoting TAPCells differentiation and lymphocyte activation. These findings provide evidence that tumor cell lysates can be functionally stabilized and even potentiated through nanoencapsulation, reinforcing the concept that delivery platforms not only preserve but also enhance antigen-driven immune responses.

## 1. Introduction

Melanoma is a highly aggressive skin cancer with steadily increasing incidence worldwide, including in Chile, where cases have tripled in the past three decades [1]. Despite significant advances in cancer immunotherapy, particularly immune checkpoint blockade (ICB), a substantial proportion of patients remain refractory or acquire resistance, underscoring the need for complementary therapeutic strategies [2,3,4]. Therapeutic cancer vaccines have re-emerged as attractive alternatives by directly priming and expanding tumor-specific T cells. Among them, whole-tumor-cell (WTC) vaccines are particularly appealing because they provide a broad antigenic repertoire, including patient-shared tumor-associated antigens [5]. However, clinical experience with WTC vaccines has been historically limited by suboptimal immunogenicity, absence of standardized adjuvancy, and heterogeneous clinical outcomes [5].

Clinically validated examples of WTC vaccines are TAPCells and TRIMELVax, whose formulations rely on TRIMEL, a heat-shocked lysate derived from three allogeneic melanoma cell lines [6,7,8,9,10]. TAPCells, a dendritic cell (DC)-like vaccine loaded with TRIMEL, have demonstrated safety and clinical benefit in phase I/II trials, inducing tumor-specific T cell responses and improving survival in melanoma and prostate cancer patients [6,7,8,10]. Interestingly, heat-shock conditioning of tumor cells induces damage-associated molecular patterns (DAMPs) that optimize antigen presentation and DC differentiation, conferring superior immunogenicity compared with conventional lysates [7]. TRIMELVax is a therapeutic vaccine composed of TRIMEL combined with the potent natural adjuvant *Concholepas concholepas* hemocyanin (CCH). Preclinical studies demonstrated that TRIMELVax promotes DC activation, enhances infiltration of effector CD4^+^ and CD8^+^ T cells into tumors, and controls the aggressive B16F10 melanoma model even in the absence of checkpoint blockade [10]. More recently, we conducted a phase I clinical trial to evaluate safety, tolerability, immunogenicity, and preliminary efficacy of TRIMELVax in patients with unresectable stage IV melanoma who had progressed after first-line anti-PD-1 therapy (NCT06556004). Nevertheless, TRIMEL requires storage at ultra-low temperatures (−80 °C), which poses significant barriers for large-scale manufacturing, distribution, and commercialization, as exemplified by the logistical challenges faced by other cell-based therapies [11].

Nanoparticle-based delivery systems represent an emerging strategy to overcome these bottlenecks [12,13]. Biodegradable polymers such as poly(lactic-co-glycolic acid) (PLGA) have been extensively used in drug delivery and vaccine development due to their biocompatibility, controlled release properties, and FDA approval [14]. Several preclinical studies have demonstrated that encapsulating tumor antigens into PLGA nanoparticles can enhance antigen stability, improve DC uptake, and increase antigen cross-presentation efficiency [15,16,17,18,19,20]. Yet, to date, no reports have addressed the nanoencapsulation of clinically validated, heat-shocked tumor lysates such as TRIMEL.

Despite these advantages, PLGA-based delivery systems also present important challenges that must be considered when formulating complex antigenic preparations such as whole tumor cell lysates. PLGA degradation generates an acidic microenvironment that can promote partial denaturation or hydrolysis of nano-encapsulated proteins, potentially altering immunogenic epitopes or promoting unwanted peptide fragmentation [21,22]. Moreover, the kinetics of antigen release from PLGA matrices—shaped by polymer composition, particle size, and degradation rate—are critical determinants of functional stability and of the balance between antigen preservation and timely availability to antigen-presenting cells [15,23]. These factors highlight that the immunological potentiation associated with nanoparticulate delivery is not intrinsic to PLGA itself but emerges only when antigen integrity and release profiles are appropriately preserved. In this context, evaluating the stability and functional performance of nanoencapsulated whole tumor cell lysates becomes essential for supporting the translational value of PLGA-based vaccine platforms.

Here, we report that PLGA nanoencapsulation of TRIMEL (NP-TRIMEL) preserves its immunogenic properties for up to 24 weeks at 4 °C, while significantly enhancing its efficiency to induce TAPCells differentiation and cytotoxic lymphocyte activation in vitro. These findings provide proof of concept that nanoencapsulation not only addresses critical limitations in tumor lysate preservation but also potentiates their immunogenicity, supporting the rationale for further preclinical development of NP-TRIMEL as a next-generation cancer vaccine candidate.

Importantly, the conceptual novelty of this study does not lie in PLGA nanoencapsulation itself, a well-established delivery platform, but rather in applying this technology to TRIMEL, a clinically validated heat-shock melanoma tumor cell lysate used in TAPCells and TRIMELVax vaccines. Stabilizing and potentially potentiating a lysate with demonstrated clinical activity constitutes the true translational innovation of this work.

## 2. Materials and Methods

Synthesis of nanoparticles: PLGA nanoparticles were synthesized using a double emulsion/solvent evaporation method adapted from Roberts et al. 2020 [24] with modifications based on Kohnepoushi et al. 2019 [18]. Briefly, 50 mg of PLGA (50:50; molecular weight 30,000–60,000, #Lot: MKCK9247; P2191; Sigma-Aldrich, Saint Louis, MO, USA) were dissolved in 1 mL of dichloromethane anhydrous (DCM; ≥99.8%, containing 40–150 ppm amylene as stabilizer; Merck, Sigma-Aldrich, Saint Louis, MO, USA) under vortex mixing. Three formulations were prepared: (1) NP-TRIMEL containing 50 μL of TRIMEL lysate (4.7 μg peptides/μL; 235 μg net peptides); (2) NP-Veh containing 50 μL of serum-free clinical grade AIM-V^®^ medium CTS^TM^ (Gibco, Invitrogen, Grand Island, NY, USA) (vehicle control); and (3) empty nanoparticles (NP-empty) with 50 μL MilliQ water. Each mixture was sonicated on ice for 2 min at 50% amplitude (QSonica CL-188, Newton, CT, USA). The primary emulsion was added dropwise into 10 mL of 2.5% (*w*/*v*) polyvinyl alcohol (PVA; 98–99% hydrolyzed, low molecular weight; CAS:9002-89-5; Alfa Aesar, Thermo Fisher Scientific, Ward Hill, MA, USA) under constant stirring. The PVA solution was prepared by dissolving PVA in MilliQ water at 80 °C for 1 h under magnetic stirring (380 rpm). A second sonication (2 min, 50% amplitude, on ice) was performed, followed by the addition of 50 mL of 0.3% (*w*/*v*) PVA and continuous stirring for 12 h to allow complete DCM evaporation. Nanoparticles were washed twice in MilliQ water by centrifugation (12,500 rpm, 30 min, 4 °C; Kubota 3520), resuspended in 10 mL MilliQ water, and distributed into 2 mL Eppendorf tubes. Quality control was assessed by dynamic light scattering (DLS), and nanoparticles were lyophilized and stored at 4 °C until use.

Nanoparticles were lyophilized for 48 h (BIOBASE BK-FD10P STALab) after prior freezing at −80 °C for 24 h (Haier Ultra Low Temperature Freezer). Samples were introduced into the lyophilizer at −60 °C with a vacuum of <10 Pa. Lyophilized nanoparticles were stored at −80 °C (baseline, T_0_) or at 4 °C for 1, 4, or 24 weeks.

DLS: 100 μL of nanoparticle suspension was diluted to 1 mL in distilled water and analyzed in a Zetasizer Nano (Malvern Instruments, Malvern, UK). Hydrodynamic diameter, polydispersity index (PdI), and zeta potential were recorded using dedicated capillary cells and analyzed with Zetasizer software v7.11. The used parameters were scattering angle, 90°; temperature, 25 °C; number of runs, 3 × 100.

Field-Emission Scanning Electron Microscopy (FE-SEM) and Energy-Dispersive X-ray Spectroscopy (EDS): Lyophilized nanoparticles were resuspended in 10 μL of water and deposited onto silicon wafers. After drying, samples were gold-coated (8 nm; Creesington Sputter Coater 108auto) and imaged with FE-SEM (GeminiSEM 360, Zeiss, Oberkochen, Germany). Elemental composition was analyzed by EDS (Oxford Ultim Max 40, Abingdon, UK) using Aztec software version 6.2.

TRIMEL-derived peptide loading determination: TRIMEL loading was quantified using the Pierce Quantitative Colorimetric Peptide Assay (ThermoFisher Scientific, 23275, Rockford, IL, USA), following manufacturer instructions with adaptations from Kohnepoushi et al. 2019 [18]. Lyophilized nanoparticles (5 mg) were dissolved in 1 mL of 1 M NaOH (PanReac AppliChem, Ottoweg, Germany) and incubated for 24 h at 37 °C under agitation (200 rpm; Shaker KS 3000i, IKA, Saint Louis, MO, USA). After centrifugation (10,000 rpm, 20 min, 4 °C; Kubota 3520, Tokyo, Japan), supernatants were collected and analyzed at 480 nm using a microplate reader (Synergy HTX, BioTek Instruments, Winooski, VT, USA).

Cell cultures and TRIMEL lysate: human melanoma cell lines Mel1, Mel2, and Mel3 (components of TRIMEL) were maintained in RPMI-1640 medium (HyClone, Logan, UT, USA) supplemented with 10% fetal bovine serum (FBS; HyClone, Logan, UT, USA) and penicillin/streptomycin (Corning, Manassas, VA, USA) at 37 °C, 5% CO_2_, as described in Aguilera et al. [7]. TRIMEL was produced as described before [7]. Briefly, Mel1, Mel2, and Mel3 cells were harvested using PBS/EDTA 0.05% (Corning, Manassas, VA, USA), washed with PBS buffer, and counted using Neubauer’s chamber. Cells were mixed in equal amounts to achieve a final concentration of 4 × 10^6^ cells/mL in serum-free clinical grade AIM-V^®^ medium CTS^TM^. The mixed cells were subjected to heat shock stress by culturing them to 42 °C for 1 h plus 37 °C for two additional hours. Then, the cells were lysed by three freeze–thawing cycles (liquid nitrogen and then at 37 °C, respectively). TRIMEL was free of endotoxins, as it was tested using PyroGene Recombinant Factor C Endotoxin Detection Assay (Lonza Group, Walkersville, MD, USA). Peripheral blood mononuclear cells (PBMCs) from three melanoma patients were isolated by leukapheresis. These three patients, two males (73 and 75 years old) and one female (36 years old), are part of a cohort of melanoma stage IV patients included in a TRIMELVax phase I clinical trial (NCT06556004). Eligible patients of this cohort were aged 18 years or older, had histologically confirmed melanoma, an Eastern Cooperative Oncology Group (ECOG) performance status of 0–1, had experienced progression or unacceptable toxicity following first-line anti-PD-1 immunotherapy and had adequate organ function. Exclusion criteria included active autoimmune disease and concurrent immunosuppressive therapy. Monocytes were differentiated into activated monocytes (AM) by culture in RPMI supplemented with rhIL-4 (500 U/mL; US Biological, Swampscott, MA, USA) and rhGM-CSF (800 U/mL; Schering Plough, Brinny Co., Cork, Ireland) for 48 h. AMs were seeded in 96-well plates (5 × 10^4^ cells/well) and stimulated with TRIMEL (20 μL/well) (94 μg of peptides/well), NP-Veh (250 μg/well), or NP-TRIMEL (250 μg/well) to generate TAPCells. Before use, lyophilized nanoparticles were resuspended in PBS at 10 mg/mL.

Flow cytometry: AMs were harvested, seeded in 48-well plates (1 × 10^5^ cells/well), and stimulated for 24 h with different concentrations (12, 25, 50, 100, 200, and 400 μg/mL) of NP-Veh or NP-TRIMEL. As positive and negative controls, AMs were stimulated for 24 h with TRIMEL (15, 60, or 120 μg peptides/mL) or left untreated, respectively. The expression of the activation marker CD80 (clone 2D10-4, BV421) was determined in the CD11c^+^ population (clone 3.9, APC) using monoclonal antibodies (eBioscience, San Diego, CA, USA). After washing with staining buffer (PBS, 3% FBS), cells were incubated with fluorochrome-conjugated antibodies for 60 min at 4 °C in the dark. Cells were then washed three times with staining buffer and analyzed on an LSRFortessa X-20 flow cytometer (BD Biosciences, Franklin Lakes, NJ, USA). Data were processed using BD FACSDiva software version 8.0.3.

TAPCells-lymphocytes co-cultures: the different TAPCells were co-cultured with autologous peripheral blood lymphocytes (PBLs) at a 1:10 ratio in the presence of IL-2 (100 U/mL) for 7 days, with medium change on day 4. PBLs were then re-stimulated with autologous TAPCells (5 × 10^4^ cells/well) for five additional days. Activated PBLs were subsequently used in cytotoxicity assays.

Cytotoxicity assays: cytotoxicity was evaluated by calcein-AM release, as described by Neri et al. (2001) [25]. Briefly, melanoma target cells (Mel1, Mel2, or Mel3) were labeled with 1 μM calcein-AM (Invitrogen, Eugene, OR, USA) for 30 min at 37 °C, washed, and seeded at 1 × 10^4^ cells/well in 96-well plates. Effector PBLs, from three independent melanoma patients, were added at an effector: target (E:T) ratio of 20:1. Fluorescence in the supernatant (Ex: 480 nm, Em: 530 nm) was measured after 24 h (Varioskan Lux, Thermo Fisher Scientific, Rockford, IL, USA). Note: As PBLs were tested against the three melanoma cell lines included in the TRIMEL, this generated nine possible donor–target combinations (3 × 3). Four combinations exhibited putative HLA compatibility, defined by detectable cytotoxic activity against at least one melanoma cell line following activation with TAPCells generated using soluble TRIMEL. These combinations were: Patient #1 → Mel2; Patient #2 → Mel3; and Patient# 3 → Mel1 and Mel2. Only these four biologically compatible combinations were included in cytotoxicity analyses. Each combination represents an independent co-culture of (i) a biologically distinct TAPCells–PBL population and (ii) an independently cultured melanoma target line.

For all comparisons between soluble TRIMEL and NP-TRIMEL, normalization was performed using the experimentally quantified amount of TRIMEL-derived peptides encapsulated in each NP batch at the corresponding storage time.

Statistical analysis: data were analyzed with GraphPad Prism 10.0 (GraphPad Software, San Diego, CA, USA). Statistical analyses were performed using one-way ANOVA followed by Tukey’s multiple-comparisons test. Biological replicates (*n*) for each assay are indicated in the respective figure legends, and results are presented as mean ± SEM. A *p*-value < 0.05 was considered statistically significant.

## 3. Results

### 3.1. Physicochemical Characterization of NP-TRIMEL

NP-TRIMEL synthesized by a double-emulsion/solvent evaporation method exhibited the expected physicochemical characteristics of PLGA nanoparticles. DLS analysis revealed an average hydrodynamic size ranging 211–256 nm (mean = 227 ± 16 nm), and a low polydispersity index (PDI; <0.1), consistent with uniform nanoparticle distribution. The zeta potential measurements indicated a negative surface charge (mean = −17 mV), supporting colloidal stability in aqueous suspension (Figure 1A,B). Control nanoparticles showed similar hydrodynamic sizes: NP-empty, 251 ± 23 nm and NP-Veh, 224 ± 28 nm (both with PDI < 0.1). The zeta potentials of control nanoparticles were slightly less than those of NP-TRIMEL nanoparticles, but still negative: NP-empty, −9 mV and NP-Veh, −8 mV. FE-SEM confirmed the spherical morphology of NP-TRIMEL, with smooth surfaces and narrow size distribution (Figure 1C). Importantly, SEM-based size quantification demonstrated that NP-TRIMEL preserved its morphology and size during storage at 4 °C for 1, 4, and 24 weeks, without significant alterations (Figure 1D). To further assess colloidal stability beyond SEM morphology, we evaluated hydrodynamic size and zeta potential of NP-TRIMEL after storage at 4 °C (Figure 1E). Although SEM-derived diameters remained unchanged, DLS revealed a progressive increase in hydrodynamic size, suggesting early-stage NP agglomeration. In parallel, NP-TRIMEL exhibited a shift toward less negative zeta potential values, indicating partial surface charge neutralization during storage. Both parameters stabilized after week 4, mirroring the plateau observed in peptide loss, as will be shown later. Importantly, these changes did not compromise NP-TRIMEL dispersibility during resuspension nor their biological activity in subsequent functional assays.

### 3.2. TRIMEL Encapsulation Efficiency and Peptide Preservation in TRIMEL-NP

Encapsulation of TRIMEL into PLGA nanoparticles was confirmed by elemental analysis. EDS mapping revealed the expected elemental composition, dominated by carbon and oxygen, consistent with the chemical structure of the lactic acid–glycolic acid copolymer. Importantly, low but detectable nitrogen signals were observed exclusively in NP-TRIMEL and NP-Veh, but not in NP-empty, supporting the presence of TRIMEL- and culture medium-derived proteins either encapsulated within or adsorbed onto the nanoparticle surface (Figure 2A,B). Additional signals for sodium, chlorine, phosphorous, potassium, and calcium were also identified, likely attributable to components present in the culture medium and in the cells from which TRIMEL was manufactured. These results confirm the expected elemental profile of PLGA nanoparticles and provide further evidence of protein incorporation into NP-TRIMEL.

Quantitative peptide assays corroborated these findings, revealing concentrations of 3.9 mg of TRIMEL-derived peptides per milligram of nanoparticle (Figure 2C, red square). Notably, TRIMEL-derived peptides remained detectable after storage for up to 24 weeks at 4 °C. Our results show that NP-TRIMEL loses approximately 40% of its initial TRIMEL load after 4 weeks at 4 °C, with no further reduction observed during the subsequent 20 weeks (Figure 2C). Nevertheless, nanoparticles stored for extended periods retained significant amounts of TRIMEL, averaging 2.2 μg of TRIMEL peptides per mg of NP.

### 3.3. Biological Activity of NP-TRIMEL

To determine whether NP-TRIMEL exhibits functional biological activity, first, we evaluate its capacity to differentiate TAPCells. AM from melanoma patients were stimulated with soluble TRIMEL, NP-TRIMEL, or control NP-Veh. Flow cytometry analysis revealed a marked upregulation of the DC activation marker CD80 in the CD11c^+^ population following NP-TRIMEL treatment, but not NP-Veh (Figure 3A,B). When normalized by the net peptide content, NP-TRIMEL displayed over a 300-fold higher efficiency than soluble TRIMEL in driving TAPCells differentiation (Figure 3B, right panel).

Then we test the preservation of TRIMEL-NP’s biological activity over different storage periods at 4 °C. To address this, we evaluated whether NP-TRIMEL contains melanoma-associated antigens that can be efficiently taken up and processed by AM, thereby generating TAPCells capable of activating lymphocytes against melanoma cells. TAPCells generated with NP-TRIMEL were co-cultured with autologous PBLs, and the resulting effector lymphocytes exhibited potent cytotoxic activity against TRIMEL melanoma cell lines, as measured by calcein-release assays (Figure 3C). Remarkably, this cytotoxic function was preserved even after 24 weeks of storage at 4 °C with NP-TRIMEL. Normalization of cytotoxic activity was then performed using the measured encapsulated TRIMEL-peptide content of each NP-TRIMEL batch at its corresponding storage time. This allowed a direct comparison of functional potency per unit peptide mass between soluble TRIMEL and NP-TRIMEL. Interestingly, when this cytotoxic activity was normalized, we observed that NP-TRIMEL was two orders of magnitude more efficient than TRIMEL (Figure 3D). Together, these findings indicate that nanoencapsulation not only preserves but also potentiates TRIMEL’s capacity to induce functional TAPCells and anti-melanoma cytotoxic lymphocytes.

## 4. Discussion

The present work demonstrates that PLGA nanoencapsulation of a clinically validated, heat-shocked melanoma cell lysate (TRIMEL) preserves its immunogenic function for at least 24 weeks at 4 °C, while markedly increasing the per-protein efficiency of TAPCells induction and downstream cytotoxic responses. These findings are concordant with a large body of evidence showing that PLGA nanoparticles protect antigens from degradation, facilitate APC uptake, and enhance cross-presentation, thereby augmenting T-cell priming relative to soluble formulations [16,17,18,19].

The physicochemical stability analyses provide important mechanistic insight into the behavior of NP-TRIMEL during storage. SEM imaging confirmed that particle morphology and dry-state diameter remained stable over 24 weeks; however, DLS revealed a progressive increase in hydrodynamic size accompanied by a shift toward a less negative zeta potential. This discrepancy is expected: SEM measures dehydrated, isolated particles, whereas DLS captures nanoparticle behavior in suspension, where subtle colloidal interactions, hydration layers, and early-stage agglomeration become detectable. The concomitant decline in surface charge suggests partial surface reorganization, a phenomenon likely facilitated by the ~40% peptide loss observed during the first 4 weeks. Loss or redistribution of surface-associated peptides decreases electrostatic repulsion, promoting reversible nanoparticle–nanoparticle interactions in solution without altering their intrinsic morphology. Notably, both hydrodynamic size and zeta potential stabilized after week 4, mirroring the plateau in peptide loss and indicating that NP-TRIMEL undergoes an initial surface-adjustment phase followed by long-term colloidal stabilization.

Elemental analysis revealed nitrogen only in NP-TRIMEL and NP-Veh (but not in empty controls), supporting the presence of protein cargo. At the same time, peptide quantification documented a controlled decline (~40%) by week 4, followed by a plateau, indicating partial cargo loss but long-term retention of immunogenic material sufficient to drive robust TAPCells activation and cytotoxic function.

The low nitrogen signal detected by EDS in NP-Veh likely reflects adsorption or incorporation of nitrogen-containing components of the AIM-V clinical-grade medium used as the aqueous phase during NP synthesis. AIM-V contains amino acids, vitamins, and other nitrogenous compounds that may interact with PLGA during emulsification. Therefore, nitrogen detection in NP-Veh does not imply TRIMEL contamination but is consistent with expected interactions between PLGA and medium constituents. Because our conclusions rely on quantitative peptide loading and functional assays rather than EDS elemental composition, this observation does not affect the interpretation of NP-TRIMEL encapsulation. Future studies using X-ray photoelectron spectroscopy could further characterize these surface phenomena.

Mechanistically, heat-shock conditioning is known to enrich tumor cell lysates in DAMPs (e.g., HMGB1, HSP70/HSP90), which cooperate with tumor-associated antigens to license DC [7,8,26,27]. It is therefore plausible that PLGA provides spatiotemporal control over the delivery of both antigenic peptides and DAMP-like signals, enhancing DC maturation (CD80 upregulation) and subsequent cytotoxic lymphocyte generation observed here. Although we did not directly quantify cross-presented peptide–MHC complexes, our results parallel prior reports showing that PLGA carriers enhance cross-presentation and T-cell responses to melanoma peptide antigens [28]. Functionally, the observation that NP-TRIMEL outperforms soluble TRIMEL by two orders of magnitude, normalized to peptide mass, suggests improved dose sparing and aligns with the general principle that particulate delivery can outperform soluble antigen for priming cytotoxic responses.

From a translational standpoint, these data directly address a long-standing bottleneck in TRIMEL-based technologies (TAPCells/TRIMELVax): the logistic burden of −80 °C storage. By preserving function at 4 °C, NP-TRIMEL offers a path to a cost-effective cold chain and potentially broader access, complementing ongoing clinical efforts with TAPCells and TRIMELVax, which have shown immunologic activity and clinical signals in melanoma. Notably, the stability profile we report at 4 °C is consistent with other PLGA vaccine systems, although ultimate shelf-life will depend on polymer composition and excipient choices. Although NP-TRIMEL is intended exclusively for ex vivo use and thus 4 °C represents the clinically relevant storage condition, evaluating the formulation at 37 °C may provide additional insights into PLGA degradation kinetics and release behavior. This represents an interesting avenue for future studies.

Our brief study has some limitations. First, the colorimetric peptide assay quantifies total peptides contained in the lysate but does not resolve identity or integrity of clinically relevant antigens, nor complete proteins and/or other complex molecules of different nature present in TRIMEL; proteomics (LC-MS/MS) of released cargo and accelerated and real-time stability are warranted. Second, accurate encapsulation vs. surface adsorption remains to be distinguished; surface protease shaving, TEM with immunogold, and differential release kinetics would clarify cargo localization. Third, we did not directly measure cross-presentation; HLA-I immunopeptidomics, flow-cytometric pMHC multimers, and ELISpot/fluoroSpot for antigen-specific T cells would mechanistically strengthen the case. Additionally, although the TRIMEL lysate used in this study is GMP-like manufactured and routinely tested endotoxin-free, NP formulations were not directly assayed for endotoxin content. NP-Veh controls showed minimal CD80 upregulation, suggesting only limited contribution from the NP matrix. Nevertheless, we cannot completely exclude low-level endotoxin-mediated effects, and follow-up work will incorporate direct endotoxin quantification and TLR4 inhibition controls. Finally, in future iterations of NP-TRIMEL, incorporating cryoprotectants such as sorbitol during lyophilization may reduce early-stage desorption of peptides, maintain a more negative zeta potential, mitigate hydrodynamic size increase, and ultimately extend colloidal and functional stability beyond 24 weeks. Such formulation refinements will be essential for scaling NP-TRIMEL toward GMP manufacturing and clinical translation.

In sum, by pairing a clinically validated heat-shocked tumor cell lysate with a well-understood PLGA carrier, this study shows that nanoencapsulation can both stabilize and potentiate tumor cell-lysate immunogenicity, yielding durable activity at 4 °C and substantially higher per-dose efficiency than the soluble comparator. These attributes, together with planned in vivo validation and GMP-minded manufacturing controls, position NP-TRIMEL as a credible next-generation WTC vaccine candidate to complement ICB and other immunotherapies in melanoma and other cancers.

## 5. Conclusions

In summary, this work demonstrates that TRIMEL, a clinically validated heat-shocked melanoma cell lysate, can be successfully nanoencapsulated into PLGA nanoparticles without loss of function and with clear gains in immunogenic potency. NP-TRIMEL maintained morphological stability, retained peptide cargo for at least 24 weeks at 4 °C, and consistently induced functional TAPCells with superior efficiency compared with soluble TRIMEL. The >100-fold dose-normalized increase in activity underscores the added value of nanoencapsulated delivery. By enabling functional storage at 4 °C, NP-TRIMEL addresses a long-standing barrier that limits the scalability of TAPCells/TRIMELVax-like vaccines. Although additional proteomic, mechanistic, and in vivo studies are needed, the present findings support NP-TRIMEL as a feasible and translationally promising platform for next-generation whole-tumor-cell cancer vaccines.

## Figures and Tables

**Figure 1 cells-14-01939-f001:**
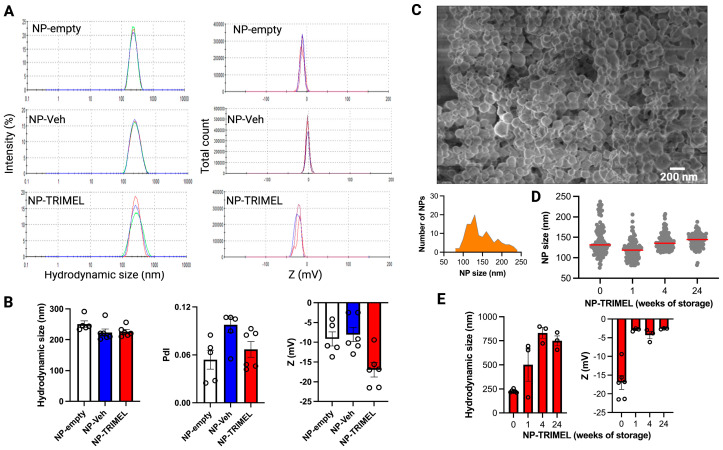
DLS, superficial charge and SEM of NP-TRIMEL. (**A**) Representative DLS histograms showing hydrodynamic sizes (left panels) and superficial charge (Z potential; right panels) of NP-empty (upper panels), NP-Veh (middle panels), and NP-TRIMEL (lower panels). (**B**) Quantification of hydrodynamic sizes, polydispersity indexes (PdI), and Z potential for different batches of nanoparticles (NP-empty, *n* = 2 batches with 2–3 replicate measurements each; NP-Veh and NP-TRIMEL, *n* = 3 batches with 2 replicate measurements each). (**C**) Representative SEM image of NP-TRIMEL nanoparticles (upper panel) and associated histogram (below). (**D**) Quantification of NP-TRIMEL nanoparticles’ size after different times of storage at 4 °C, determined by analysis of SEM images. Data represent measurements from two representative images for the three independent nanoparticle batches. Red lines indicate the median values. (**E**) Quantification of hydrodynamic sizes and Z potential for the three different batches of NP-TRIMEL after different times of storage at 4 °C.

**Figure 2 cells-14-01939-f002:**
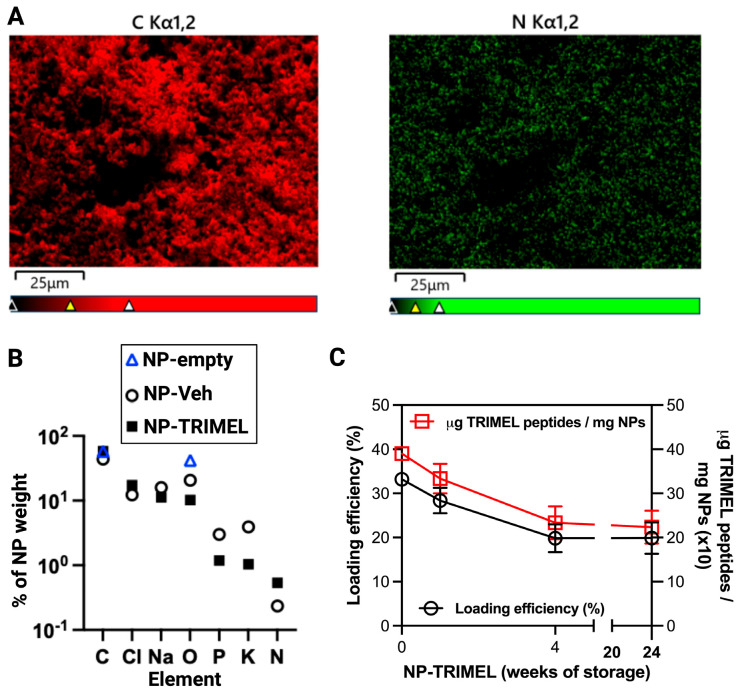
Elemental analysis and TRIMEL loading efficiency of nanoparticles. (**A**) Representative EDS mapping images of NP-TRIMEL nanoparticles for carbon (red) and nitrogen (green). (**B**) EDS quantification of elements present in nanoparticles (NP). The average value of three independent measures is shown. (**C**) Determination of the quantity of TRIMEL-derived peptides (red squares) and efficiency of loading (black circles) in nanoparticles after different times of storage at 4 °C. Data represent two measurement replicates from the three independent nanoparticle batches.

**Figure 3 cells-14-01939-f003:**
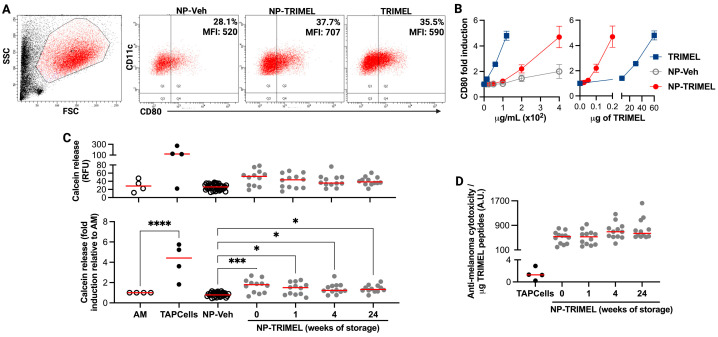
Nanoencapsulation increases the efficacy of the biological activity of TRIMEL regardless of the time of its storage at 4 °C. To induce TAPCells differentiation, activated monocytes (AM) were incubated for 24 h with varying concentrations of TRIMEL, NP-TRIMEL, NP-Veh, or maintained untreated. Expression of the dendritic cell activation marker CD80 was assessed within the CD11c^+^ population by flow cytometry. (**A**) Representative dot plots for AM stimulated with 400 μg/mL of nanoparticles or 120 μg/mL of TRIMEL, indicating the % of positive cells and the mean fluorescence intensity for CD80. (**B**) The graph on the left depicts CD80 expression in CD11c^+^ cells under different concentrations of TRIMEL (μg/mL of peptides) in the different nanoparticles or soluble lysate, expressed as fold induction relative to untreated AM (*n* = 3). The graph on the right depicts CD80 expression in CD11c^+^ cells normalized by the net quantity of TRIMEL (μg of peptides). (**C**,**D**) Immunogenic functional characterization of TRIMEL-NP after different times of storage at 4 °C. Anti-melanoma cytotoxicity assays based on calcein release. Cytotoxicity data correspond to four fully independent biological combinations, each derived from a distinct melanoma patient. PBLs from three melanoma donors were activated with autologous TAPCells generated under the indicated conditions and subsequently tested against the three melanoma cell lines composing TRIMEL (Mel1, Mel2, Mel3). From the nine possible donor–target combinations, four showed putative HLA compatibility (Patient 1 → Mel2; Patient 2 → Mel3; Patient 3 → Mel1 and Mel2) and only these combinations are shown. Each data point therefore represents an independent TAPCells–PBL co-culture challenged against an independent melanoma culture. Additionally, each point represents results for three independent batches of nanoparticles. The effector:target ratio was 20:1. TRIMEL was used at 94 μg peptides/well (376 μg/mL); NP-TRIMEL and NP-Veh at 250 μg/well (1 mg/mL). (**C**) Calcein levels measured in supernatants from co-cultures of melanoma cells with peripheral blood lymphocytes previously co-cultured with autologous antigen-presenting cells stimulated under the indicated conditions (upper panel). Results are represented as fold induction relative to the AM sample (lower panel). (**D**) Cytotoxic activity normalized to TRIMEL peptide content in both NP-TRIMEL and TRIMEL. A.U.: arbitrary units. * *p* < 0.05; *** *p* < 0.001; **** *p* < 0.0001.

## Data Availability

All published data and material are available upon reasonable request.

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
