# Peer review of "PLGA Nanoencapsulation Enhances Immunogenicity of Heat-Shocked Melanoma Tumor Cell Lysates"

_cells, 2025, doi:10.3390/cells14241939_

Round 1
Reviewer 1 Report
Comments and Suggestions for Authors
In this paper, Matheu and colleagues address the increased immunogenicity and stability of heat shocked melanoma tumor cell lysates encapsulated in PLGA. Although the report is interesting and well structured, its novelty is somewhat limited, as previous studies have already shown enhanced immune responses for different PLGA encapsulated vaccine formulations.
Therefore some concerns need to be addressed before recommendation for publication in my opinion.
Figure 1.
Please clarify in the figure legend whether the graphs in panel B represent data from a single nanoparticle batch with 5–6 independent replicate measurements, or if they reflect values obtained from independent batches with 2 replicate measurements each.
The same specification is needed for point D, the presented results are from multiple images of the same batch or different batches of NP were considered.
Figure 2.
Please specify in the figure-legend for panel C whether the peptide-loading determination was performed on a single NP batch with multiple independent measurements, or on different NP lots.
Figure 3:
Please include a representative dot plot for the TRIMEL only treatment at point A. Since this condition serves as a control throughout the figure, adding it here would improve consistency.
Why did the authors use different concentrations for the TRIMEL control (15,60, 120µg/mL) and the NP-VH/NP-TRIMEL treatments (12,25, 50, 100,200 and 400 µg/mL) when stimulating activated monocytes to generate TAPCells.
Please provide additional details in the figure legends for panels C and D regarding the experimental setup. Specifically, indicate whether the measurements shown in the graphs come from fully independent experiments involving different melanoma cultures and different batches of peripheral blood lymphocytes (PBLs) activated under the stated conditions, or if they represent only different PBL batches.
Please specify the concentration of TRIMEL , NP-VH/NP-TRIMEL used for the cytotoxicity assay presented in panel C and D.
Author Response
In this paper, Matheu and colleagues address the increased immunogenicity and stability of heat shocked melanoma tumor cell lysates encapsulated in PLGA. Although the report is interesting and well structured, its novelty is somewhat limited, as previous studies have already shown enhanced immune responses for different PLGA encapsulated vaccine formulations.
Therefore, some concerns need to be addressed before recommendation for publication in my opinion.
Figure 1: Please clarify in the figure legend whether the graphs in panel B represent data from a single nanoparticle batch with 5–6 independent replicate measurements, or if they reflect values obtained from independent batches with 2 replicate measurements each.
The same specification is needed for point D, the presented results are from multiple images of the same batch or different batches of NP were considered.
Figure 2: Please specify in the figure-legend for panel C whether the peptide-loading determination was performed on a single NP batch with multiple independent measurements, or on different NP lots.
Figure 3: Please include a representative dot plot for the TRIMEL only treatment at point A. Since this condition serves as a control throughout the figure, adding it here would improve consistency.
Why did the authors use different concentrations for the TRIMEL control (15, 60, 120 µg/mL) and the NP-VH/NP-TRIMEL treatments (12, 25, 50, 100, 200, and 400 µg/mL) when stimulating activated monocytes to generate TAPCells.
Response: We thank the reviewer for raising all these important points. The legends of Figures 1 and 2 were revised accordingly, and a representative dot plot for the TRIMEL-only condition has now been included in the new Figure 3A.
Regarding the rationale behind using different concentration ranges for TRIMEL and NP formulations: The TRIMEL concentrations were selected based on our previous experience with TRIMEL-induced TAPCell differentiation (Aguilera et al., 2011 and subsequent studies). We therefore used 120 µg/mL as the optimal reference concentration and generated two additional conditions corresponding to approximately 50% (60 µg/mL) and 10% (15 µg/mL) of this dose. In contrast, the NP concentrations were chosen according to reference values commonly used for PLGA-based nanoparticles loaded with tumor lysates in similar immunological assays. Reported working concentrations range from 100 to 500 µg/mL, including: Berti et al. (2022) (500 µg/mL), Kohnepoushi et al. (2019) (500 µg/mL), Iranpour et al. (2016) (100 µg/mL), and Prasad et al. (2011) (250 µg/mL). Based on these precedents, we selected 400 µg/mL as the upper limit—close to the average of the published concentrations—and performed subsequent 50% serial dilutions to obtain the full experimental range.
Please provide additional details in the figure legends for panels C and D regarding the experimental setup. Specifically, indicate whether the measurements shown in the graphs come from fully independent experiments involving different melanoma cultures and different batches of peripheral blood lymphocytes (PBLs) activated under the stated conditions, or if they represent only different PBL batches.
Please specify the concentration of TRIMEL, NP-VH/NP-TRIMEL used for the cytotoxicity assay presented in panel C and D.
Response: We thank the reviewer for this important clarification request. The experiments shown in Figure 3C and 3D were performed using fully independent biological samples. Peripheral blood lymphocytes (PBLs) from three melanoma patients were activated with autologous antigen-presenting cells (TAPCells) generated under the indicated conditions. These PBLs were then tested against the three melanoma cell lines that compose TRIMEL (Mel1, Mel2, Mel3).
From the nine possible combinations (3 donors × 3 melanoma lines), four exhibited putative HLA compatibility based on the ability of TAPCells generated with conventional TRIMEL to induce cytotoxic activity.
Only these four biologically compatible combinations were included in panels C and D:
Patient 1 → Mel2; Patient 2 → Mel3; Patient 3 → Mel1 and Mel2
Thus, each data point represents an independent TAPCells–PBL co-culture derived from a different patient, evaluated against an independent melanoma target culture. In the case of TAPCells generated with NPs, we also used the three independent batches of these (therefore are 12 points). We have now included this information in the figure legend and in the Methods section. Also, we included the specific concentration of TRIMEL, and the NPs used in these experiments.
Reviewer 2 Report
Comments and Suggestions for Authors
CELLS
Manuscript Number: Cells-3981507
Manuscript Title: PLGA Nanoencapsulation Enhances Immunogenicity of Heat-Shocked Melanoma Tumor Cell Lysates
Decision: Major Revisions
The manuscript entitled “PLGA Nanoencapsulation Enhances Immunogenicity of Heat-Shocked Melanoma Tumor Cell Lysates” evaluates the nanoencapsulation of TRIMEL into PLGA nanoparticles (NP-TRIMEL) as an approach to improve stability and to maintain and enhance in vitro immunogenicity (TAPCell differentiation and lymphocyte cytotoxicity). It includes a satisfactory physicochemical characterization with quantification of peptide loading and functional assays. However, several points still need to be addressed.
Q1. The Introduction section provides solid context, clearly presenting the melanoma landscape, the limitations of immune checkpoint blockade (ICB), and the rationale for therapeutic vaccines, with appropriate and actual references. However, the review of tumor-lysate antigens delivered with PLGA lacks critical notes on risks and artifacts, for example, PLGA’s acidic microenvironment and potential protein degradation, and on release kinetics as a prerequisite for functional stability, which are essential to support the hypothesis of “potentiation” by particulate delivery.
Q2. The authors should explicitly state that the novelty of this work lies in the use of the clinically validated TRIMEL (a heat-shock-treated melanoma tumor cell lysate) rather than in PLGA encapsulation.
Q3. The authors did not perform a Limulus amebocyte lysate (LAL) assay or include controls using polymyxin B or a toll-like receptor 4 (TLR4) inhibitor. Consequently, the observed upregulation of CD80 could, at least in part, be attributable to endotoxin or other contaminants rather than to the encapsulated TRIMEL.
Q5. Because PLGA carriers can activate APCs via particle properties, residual PVA and solvent, protein corona, and acidic degradation, an identically prepared, dose-matched NP-empty is required in all bioassays to attribute effects specifically to TRIMEL rather than to the nanoparticle matrix.
Q6. The claimed increase of two orders of magnitude depends on normalization to peptide content. The authors used 250 ug of nanoparticles (NPs) per well and approximately 97 ug of soluble TRIMEL per well under different conditions. It is critical to specify precisely the normalization basis (encapsulated and/or released fraction; effective recovery) and to report absolute values as well (e.g., % lysis).
Q7. Stability over 24 weeks was inferred from morphology by scanning electron microscopy (SEM) and total peptide content. It is important to include additional assays, such as dynamic light scattering (DLS), post-storage zeta potential, and in vitro release profiles, to substantiate colloidal and functional stability. This data would better support the stability claim and help explain why the load decreases by approximately 40% in 4 weeks.
Q8. It is necessary to add missing or ambiguous information, such as the number of donors used in each assay, biological variability, inclusion and exclusion criteria, and basic demographics (sex and age), as well as the measurement parameters for dynamic light scattering (DLS) (scattering angle, temperature, and number of runs).
Q9. The composition and preparation of the lysate are described only by citation. Given the central role of this research, the authors should include a concise operational summary of the methods, including the tumor cell lines used, the heat-shock protocol (temperatures, durations, and number of cycles), protein quantification procedures, and endotoxin control for the lysate (e.g., LAL assay).
Q10. Why did the authors not apply an analysis of variance (ANOVA) with an appropriate multiple-comparisons correction (e.g., Tukey’s, Holm-Sidak, or Benjamini-Hochberg)? This would be the more appropriate choice in this setting. It is also necessary to report the number of biological donors (n) for each panel/assay.
Q11. The authors should distinguish surface adsorption from true internal encapsulation, given that EDS in scanning electron microscopy is relatively insensitive for nitrogen detection; moreover, nitrogen was also observed in the vehicle-control nanoparticles. X-ray photoelectron spectroscopy (XPS) would be more informative for assessing the surface layer and composition.
Q12. The authors should correct the caption of Figure 3. The phrase “The graph on the left…” is duplicated, and ensure that all figures report scale bars, units, and the number of biological replicates (n).
Q13. Please report PLGA specifications, supplier, etc., PVA Mw/degree of hydrolysis, and solvent grades to ensure reproducibility.
Q14. Complement stability tests with 37°C (physiological condition).
Author Response
The manuscript entitled “PLGA Nanoencapsulation Enhances Immunogenicity of Heat-Shocked Melanoma Tumor Cell Lysates” evaluates the nanoencapsulation of TRIMEL into PLGA nanoparticles (NP-TRIMEL) as an approach to improve stability and to maintain and enhance in vitro immunogenicity (TAPCell differentiation and lymphocyte cytotoxicity). It includes a satisfactory physicochemical characterization with quantification of peptide loading and functional assays. However, several points still need to be addressed.
Q1. The Introduction section provides solid context, clearly presenting the melanoma landscape, the limitations of immune checkpoint blockade (ICB), and the rationale for therapeutic vaccines, with appropriate and actual references. However, the review of tumor-lysate antigens delivered with PLGA lacks critical notes on risks and artifacts, for example, PLGA’s acidic microenvironment and potential protein degradation, and on release kinetics as a prerequisite for functional stability, which are essential to support the hypothesis of “potentiation” by particulate delivery.
Response: We thank the reviewer for this valuable observation. We agree that discussing limitations associated with PLGA delivery systems is essential for providing a balanced context. We have now incorporated a new paragraph in the Introduction addressing these risks and mechanistic considerations, with appropriate references.
Q2. The authors should explicitly state that the novelty of this work lies in the use of the clinically validated TRIMEL (a heat-shock-treated melanoma tumor cell lysate) rather than in PLGA encapsulation.
Response: We thank the reviewer for this clarification. We agree that the main novelty of our study arises not from the use of PLGA NPs per se, but from the nanoencapsulation of TRIMEL, a clinically validated, heat-shock-conditioned melanoma cell lysate that underlies TAPCells and TRIMELVax vaccines. We have now added an explicit statement in the Introduction acknowledging that the innovative aspect of our work is the application of nanoencapsulation to a clinically proven tumor cell-lysate platform, rather than the PLGA methodology. This clarification emphasizes the translational relevance of stabilizing and potentiating a validated lysate used in human cancer immunotherapy.
In addition, following the editor’s guidance, we have added a Highlights section before the Abstract. These highlights succinctly summarize the main findings and their implications, emphasizing the translational significance of stabilizing and potentiating a clinically validated tumor cell lysate.
Q3. The authors did not perform a Limulus amebocyte lysate (LAL) assay or include controls using polymyxin B or a toll-like receptor 4 (TLR4) inhibitor. Consequently, the observed upregulation of CD80 could, at least in part, be attributable to endotoxin or other contaminants rather than to the encapsulated TRIMEL.
Response: We thank the reviewer for raising this important point. TRIMEL lysate used in all experiments is manufactured under GMP-like conditions and is routinely tested for endotoxin using the PyroGene Recombinant Factor C Endotoxin Detection Assay (Lonza). TRIMEL batch employed in this study passed endotoxin testing within the acceptable limits for clinical-grade materials (we now included in Methods a description of TRIMEL generation). Regarding the NPs, although NP batches were prepared under sterile conditions, we acknowledge that we did not perform an endotoxin assay directly on the final NP formulations. Nevertheless, the NP-Veh control (vehicle-loaded NPs) induced minimal CD80 upregulation compared with NP-TRIMEL (Figure 3B, grey line), indicating that the particulate carrier alone—whether via potential low-level endotoxin traces or intrinsic particle effects—accounts for only a small fraction of the observed activation. Importantly, the robust CD80 induction seen with NP-TRIMEL is substantially higher than that elicited by NP-Veh, supporting that the response is primarily driven by TRIMEL-derived immunogenic components rather than by endotoxin contamination. However, we agree with the reviewer that a contribution from residual endotoxin in NPs cannot be formally excluded, and we have now acknowledged this in the Discussion.
Q5. Because PLGA carriers can activate APCs via particle properties, residual PVA and solvent, protein corona, and acidic degradation, an identically prepared, dose-matched NP-empty is required in all bioassays to attribute effects specifically to TRIMEL rather than to the nanoparticle matrix.
Response: We agree that appropriate NP controls are essential to distinguish TRIMEL-specific effects from potential activation induced by PLGA carriers, residual PVA, surface-adsorbed proteins, or acidic degradation products. In the biological assays we systematically included NP-Veh, a control NP formulation synthesized identically to NP-TRIMEL but loaded only with vehicle (AIM-V medium). NP-Veh therefore controls for all variables of the NP matrix, including polymer composition, PVA, solvent residues, surface chemistry, and protein corona formation, except for the presence of TRIMEL cargo. As shown in Figure 3B, NP-Veh induces only minimal CD80 upregulation, confirming that the PLGA matrix alone has a very limited impact on TAPCells activation under our experimental conditions. In contrast, NP-TRIMEL triggers a markedly higher response, supporting that the observed effect is driven primarily by TRIMEL-derived immunogenic components.
Finally, although NP-empty (NPs loaded with sterile water) was synthesized and characterized physicochemically, NP-Veh was selected as the most appropriate biological control because it more faithfully reproduces all non-TRIMEL variables present in NP-TRIMEL, providing a more conservative and stringent comparison.
Q6. The claimed increase of two orders of magnitude depends on normalization to peptide content. The authors used 250 ug of nanoparticles (NPs) per well and approximately 94 ug of soluble TRIMEL per well under different conditions. It is critical to specify precisely the normalization basis (encapsulated and/or released fraction; effective recovery) and to report absolute values as well (e.g., % lysis).
Response: We thank the reviewer for this insightful comment. The normalization used to compare soluble TRIMEL and NP-TRIMEL was performed using the experimentally quantified TRIMEL-derived peptides encapsulated in each NP-TRIMEL batch, and at each storage time, as determined with the Pierce Quantitative Colorimetric Peptide Assay. In other words, the TRIMEL peptide mass contributed by NP-TRIMEL in each assay was calculated based on the actual encapsulated peptide content of the specific batch and time point used, rather than using a fixed encapsulation value. This approach accounts for batch-to-batch variability and for the expected decrease in peptide loading during long-term storage. Soluble TRIMEL was used at its experimentally measured peptide concentration (94 µg per well), allowing a direct comparison of immunogenic efficiency per unit of TRIMEL peptides delivered. Using this normalization strategy, NP-TRIMEL consistently induced comparable or superior TAPCells activation and lymphocyte-mediated cytotoxicity while delivering approximately two orders of magnitude less TRIMEL peptide mass than soluble TRIMEL. We have now clarified this normalization procedure in the Methods and Results sections.
Regarding absolute cytotoxicity values, our assay was designed around calcein-AM release normalized to untreated activated monocytes (AM) controls, as is standard in the absence of a detergent-based 100% lysis control. Because lytic-buffer conditions can interfere with fluorescence baselines—we elected not to include a maximum-lysis condition. Nevertheless, we now provide in the new version of Figure 3C (upper panel) the raw data for the relative fluorescence units (RFU) obtained and used to calculate the fold-induction relative to AM (lower panel).
Q7. Stability over 24 weeks was inferred from morphology by scanning electron microscopy (SEM) and total peptide content. It is important to include additional assays, such as dynamic light scattering (DLS), post-storage zeta potential, and in vitro release profiles, to substantiate colloidal and functional stability. This data would better support the stability claim and help explain why the load decreases by approximately 40% in 4 weeks.
Response: We thank the reviewer for this important suggestion. In the revised version, we incorporated additional physicochemical stability analyses to complement SEM morphology and peptide content. Specifically, we now provide DLS and zeta potential measurements for NP-TRIMEL after storage at 1, 4 and 24 weeks at 4 °C (new Figure 1E). These data reveal that, although SEM morphology remains unchanged, NP-TRIMEL exhibits a progressive increase in hydrodynamic diameter and a shift toward a less negative zeta potential, both consistent with incipient NP agglomeration during storage. Importantly, these changes stabilize after week 4, mirroring the plateau observed in TRIMEL peptide loss. Despite these colloidal changes, NP-TRIMEL retained full functional activity, as evidenced by preserved TAPCells activation and cytotoxic lymphocyte induction across all storage periods. Thus, the new DLS and Z potential data support a scenario where partial peptide release and mild surface reorganization occur during early storage, but without compromising NP integrity or biological performance. We discuss these points in the revised Results and Discussion sections.
Q8. It is necessary to add missing or ambiguous information, such as the number of donors used in each assay, biological variability, inclusion and exclusion criteria, and basic demographics (sex and age), as well as the measurement parameters for dynamic light scattering (DLS) (scattering angle, temperature, and number of runs).
Response: All the solicited information was included in the Methods and Results sections. We appreciate this important suggestion.
Q9. The composition and preparation of the lysate are described only by citation. Given the central role of this research, the authors should include a concise operational summary of the methods, including the tumor cell lines used, the heat-shock protocol (temperatures, durations, and number of cycles), protein quantification procedures, and endotoxin control for the lysate (e.g., LAL assay).
Response: All the solicited information was included in the Methods section. We appreciate this important suggestion.
Q10. Why did the authors not apply an analysis of variance (ANOVA) with an appropriate multiple-comparisons correction (e.g., Tukey’s, Holm-Sidak, or Benjamini-Hochberg)? This would be the more appropriate choice in this setting.
Response: We effectively applied ANOVA with Tukey’s correction for multiple comparisons (as shown in the next table). It was an unvoluntary mistake in the redaction of the Methods section. This was corrected appropriately. We appreciate this observation.
It is also necessary to report the number of biological donors (n) for each panel/assay.
We thank the reviewer for this important clarification request. The experiments shown in Figure 3C and 3D were performed using fully independent biological samples. Peripheral blood lymphocytes (PBLs) from three melanoma patients were activated with autologous TAPCells generated under the indicated conditions. These PBLs were then tested against the three melanoma cell lines that compose TRIMEL (Mel1, Mel2, Mel3).
From the nine possible combinations (3 donors × 3 melanoma lines), four exhibited putative HLA compatibility based on the ability of TAPCells generated with conventional TRIMEL to induce cytotoxic activity.
Only these four biologically compatible combinations were included in panels C and D: Patient 1 → Mel2; Patient 2 → Mel3; Patient 3 → Mel1 and Mel2.
Thus, each data point represents an independent TAPCells–PBL co-culture derived from a different patient, evaluated against an independent melanoma target culture. In the case of TAPCells generated with NPs, we also used the three independent batches of these (therefore are 12 points). We have now included this information in the figure legend and in the Methods section.
The Figure 1 and 2 legends were also improved incorporating these recommendations.
Q11. The authors should distinguish surface adsorption from true internal encapsulation, given that EDS in scanning electron microscopy is relatively insensitive for nitrogen detection; moreover, nitrogen was also observed in the vehicle-control nanoparticles. X-ray photoelectron spectroscopy (XPS) would be more informative for assessing the surface layer and composition.
Response: We thank the reviewer for this technically important point. We agree that N detection by EDS cannot discriminate between surface adsorption and internal encapsulation, and that the presence of N in NP-Veh deserves clarification. NP-Veh was prepared using the same protocol and the same aqueous phase as NP-TRIMEL, with the only difference being the absence of TRIMEL. Importantly, the vehicle used for NP-Veh is AIM-V medium, which contains several N-bearing components (e.g., amino acids, vitamins, and other small molecules from the hydrolysate fraction). Adsorption or entrapment of such compounds—expected in any water-in-oil-in-water PLGA double-emulsion system—can therefore account for the low N signal observed in NP-Veh. Thus, the N detected in NP-Veh does not represent evidence of unwanted TRIMEL contamination but rather reflects expected incorporation/adsorption of N-containing constituents of AIM-V culture medium during NP formation.
Although EDS has limitations in detecting N and cannot discriminate between surface-bound and internally encapsulated species, our conclusions do not rely on EDS as definitive evidence of encapsulation. Importantly, the distinction between surface adsorption and true encapsulation was addressed in our study through the quantitative peptide assay performed after complete PLGA dissolution, which provides direct evidence of internal cargo. In this assay, NPs were dissolved in 1 M NaOH for 24 h, fully degrading the polymer and releasing the entrapped material. The measured TRIMEL-derived peptide content cannot be explained by surface adsorption alone, since surface-bound proteins/peptides would be removed during the multiple centrifugation/washing steps performed prior to NP lyophilization. This method is widely used in PLGA literature to specifically quantify internally encapsulated peptides or proteins. Nevertheless, we agree that XPS could provide more detailed surface-composition information, and we now mention this as an interesting avenue for future mechanistic work.
Q12. The authors should correct the caption of Figure 3. The phrase “The graph on the left…” is duplicated, and ensure that all figures report scale bars, units, and the number of biological replicates (n).
Response: We modified figure legends according to these indications.
Q13. Please report PLGA specifications, supplier, etc., PVA Mw/degree of hydrolysis, and solvent grades to ensure reproducibility.
Response: All this information was incorporated in Methods section.
Q14. Complement stability tests with 37°C (physiological condition).
Response: We thank the reviewer for this suggestion. To date, NP-TRIMEL is not intended for systemic administration nor for in vivo circulation at physiological temperature. Instead, NP-TRIMEL is intended to be used ex vivo to activate autologous monocytes and generate TAPCells prior to reinfusion. For this reason, the clinically relevant stability parameter for this platform is long-term storage at 4°C, the temperature at which the product is handled and stored in a GMP-like workflow. Nevertheless, we agree that evaluating NP-TRIMEL behavior at physiological temperature is scientifically interesting, particularly to understand polymer dynamics and release kinetics under accelerated degradation conditions. As such, we have added this point to the Discussion and consider it a valuable direction for future work, especially for exploring potential alternative applications of NP-TRIMEL (for example in technologies as TRIMELVax) or for characterizing the system under stress-testing conditions.
We appreciate the reviewer’s comment, which helped us clarify the rationale behind our chosen stability conditions and identify a relevant future line of research.

Round 2
Reviewer 2 Report
Comments and Suggestions for Authors
CELLS
Manuscript Number: Cells-3981507
Manuscript Title: PLGA Nanoencapsulation Enhances Immunogenicity of Heat-Shocked Melanoma Tumor Cell Lysates
Decision: Minor Revisions
Q1. To substantiate the claim of functional stability and to interpret the ~40% decrease in loading by week 4, please provide cumulative in vitro release profiles at pH 7.4 (physiological) and pH 5.5-6.0 (endosomal mimic). Include time and simple kinetic fits to support mechanistic discussion.
Q2. Beyond NP-Veh, include a dose-matched NP-empty arm (identical polymer mass, surfactant, washing, and storage). This control is essential to isolate antigen-specific effects from those potentially driven by the nanoparticle matrix (particle uptake, surface charge, residual PVA/solvent, acidic degradation products).
Author Response
Q1. To substantiate the claim of functional stability and to interpret the ~40% decrease in loading by week 4, please provide cumulative in vitro release profiles at pH 7.4 (physiological) and pH 5.5-6.0 (endosomal mimic). Include time and simple kinetic fits to support mechanistic discussion.
Response: We thank the reviewer for this valuable suggestion. We fully agree that detailed in vitro release profiling under physiological (pH 7.4) and endosomal-mimicking conditions (pH 5.5–6.0), together with kinetic modeling, would provide deeper mechanistic insight into the early 40% decrease in peptide load. However, generating these datasets would require a new experimental series that cannot be incorporated at this stage of the revision.
While we cannot add new release experiments, the current dataset already provides some evidence supporting our interpretation of the early loss phase and the long-term functional stability:
The plateau in TRIMEL peptide content after week 4 strongly suggests that the initial 40% decrease represents a desorption/redistribution phase rather than progressive degradation.
The parallel stabilization of hydrodynamic diameter and Z potential mirrors the plateau in TRIMEL peptide content, indicating that both phenomena likely reflect early surface reorganization rather than continued peptide release or degradation.
The maintained immunogenic function at weeks 1, 4, and 24 demonstrates that the loss of 40% of peptide mass does not compromise functional stability.
We hope the reviewer finds this clarification satisfactory.
Q2. Beyond NP-Veh, include a dose-matched NP-empty arm (identical polymer mass, surfactant, washing, and storage). This control is essential to isolate antigen-specific effects from those potentially driven by the nanoparticle matrix (particle uptake, surface charge, residual PVA/solvent, acidic degradation products).
Response: We thank the reviewer for raising this methodological point. However, we respectfully disagree that a dose-matched NP-empty control would provide a more appropriate negative control than NP-Veh for the specific biological question addressed in this study. NP-Veh contains the same vehicle solution used to formulate soluble TRIMEL, i.e., serum-free AIM-V clinical-grade medium. This is critical because: AIM-V does not induce monocyte activation or differentiation, and NP-Veh recreates the full physicochemical context of NP-TRIMEL except for the presence of the cell lysate TRIMEL. Thus, NP-Veh controls for all potential contributions of: PLGA polymer, PVA surfactant, residual solvent, colloidal behavior, storage effects, and the presence of culture medium components carried through the aqueous pase.
Likewise, physicochemical profiles of NP-Veh and NP-empty are nearly identical, showing minimal differences attributable to the internal aqueous phase. Together, these results indicate that the PLGA matrix itself does not drive APC activation, and that the strong immunogenic responses observed with NP-TRIMEL are due specifically to TRIMEL content, not to PLGA-associated artifacts.
Nevertheless, although dose-matched NP-empty NPs were synthesized and included in physicochemical analyses, patient-derived monocytes were limiting, preventing their parallel use in functional assays. NP-Veh was therefore selected as the most appropriate and biologically sound control.